# A Proposal for Streamlining 3D Digital Cadastral Data Lifecycle

Hamed Olfat * , Behnam Atazadeh , Farshad Badiee, Yiqun Chen, Davood Shojaei and Abbas Rajabifard

The Centre for Spatial Data Infrastructures and Land Administration, Department of Infrastructure Engineering, The University of Melbourne, Melbourne, VIC 3010, Australia; behnam.atazadeh@unimelb.edu.au (B.A.); farshad.badiee@unimelb.edu.au (F.B.); yiqun.c@unimelb.edu.au (Y.C.); shojaeid@unimelb.edu.au (D.S.); abbas.r@unimelb.edu.au (A.R.)
* Correspondence: olfath@unimelb.edu.au

**Abstract:** In urban areas, managing the lifecycle of land and property data related to interlocked and intertwined structures and infrastructure services is a grand challenge for cadastral systems. Addressing the physical and legal complexities of vertically stratified ownership arrangements is a major step towards the modernization of cadastral systems. The research problem that this study addresses is the lack of a simplified and effective approach for modelling, storing, visualizing, and querying 3D cadastral data for multi-story buildings. This research primarily leads to the development of an approach based on Building Information Modelling (BIM), as well as state-of-the-art ETL (extract, transform, load), database and visualization technologies for 3D cadastral data lifecycle management in current practices. The proposed steps for recording, preserving, and disseminating 3D cadastral data are crucial in shifting current 2D cadastral systems towards 3D digital information systems. The results showed improvements in data creation, storage, conversion, and communication when upgrading from a 2D to 3D digital cadastre. Therefore, this study confirmed that streamlining the lifecycle of cadastral data using 3D environments would mitigate issues associated with the current fragmented 2D cadastral datasets used in the multi-story developments.

**Keywords:** 3D digital cadastre; data lifecycle; 3D legal spaces; BIM; 3D Tiles

## 1. Introduction

There are many interlocked and interconnected buildings and infrastructure facilities in urban environments. This complexity creates a significant challenge for managing land and property information in these environments. Addressing the spatial and legal complexity of multi-layered ownership situations is a significant leap in realizing modern cadastral systems. A 3D digital cadastre, which aims at streamlining the recording, management and dissemination of complex rights, restrictions, and responsibilities (RRRs) using 3D spatial information models, has been one of the major research and developmental areas in different countries, such as Australia [1,2], Canada [3], China [4], Czech [5], Malaysia [6,7], New Zealand [8], Singapore [9], Sweden [10–12], and the Netherlands [13,14]. However, there is not a fully functional 3D cadastral platform in any jurisdiction yet due to the complexities of implementing such a platform. The complexities can arise from technical, institutional, and legal aspects of the 3D digital cadastre [15]. Combined complexities arising from these aspects are also possible. For instance, while complexities of technical solutions, as presented in this study, should be addressed for 3D digital cadastre, the legal complexity of adopting 3D digital models according to current regulations and acts should be considered in parallel. There is a wide range of investigations that have explored these aspects.

The technical aspect of the 3D digital cadastre mainly includes the general steps for a digital data lifecycle shown in Figure 1. The data preparation consists of data collection methods [16,17], data modelling [18,19], validation of data [20], and updating the data when required. Data query and analysis is the other main step of the digital data lifecycle

which enables the land administration stakeholders to identify and examine the model components and make informed decisions [21]. Each of the steps represented in Figure 1 plays a critically important role in implementing digital cadastral systems. This implementation should start with capturing and preparing digital data with high quality and reliability. Once the digital data are prepared, they need to be stored as a flat file or in a database. Spatial data models provide the essential elements for managing cadastral objects and their relationships. There is a wide range of spatial data models, such as the Land Administration Domain Model (LADM), Land and Infrastructure (LandInfra), CityGML, Industry Foundation Classes (IFC), and combinations of these data models. Selecting an appropriate data model depends on specific digital data requirements in each jurisdiction. Effectively stored digital cadastral data need to be visualized, queried, and analyzed in a desktop or web/mobile application. Good mechanisms for visual communication of digital cadastral data and retrieving the on-demand knowledge about this information will unlock its value for a range of key stakeholders, such as urban planners, land registries, and property owners involved in complex land development processes.

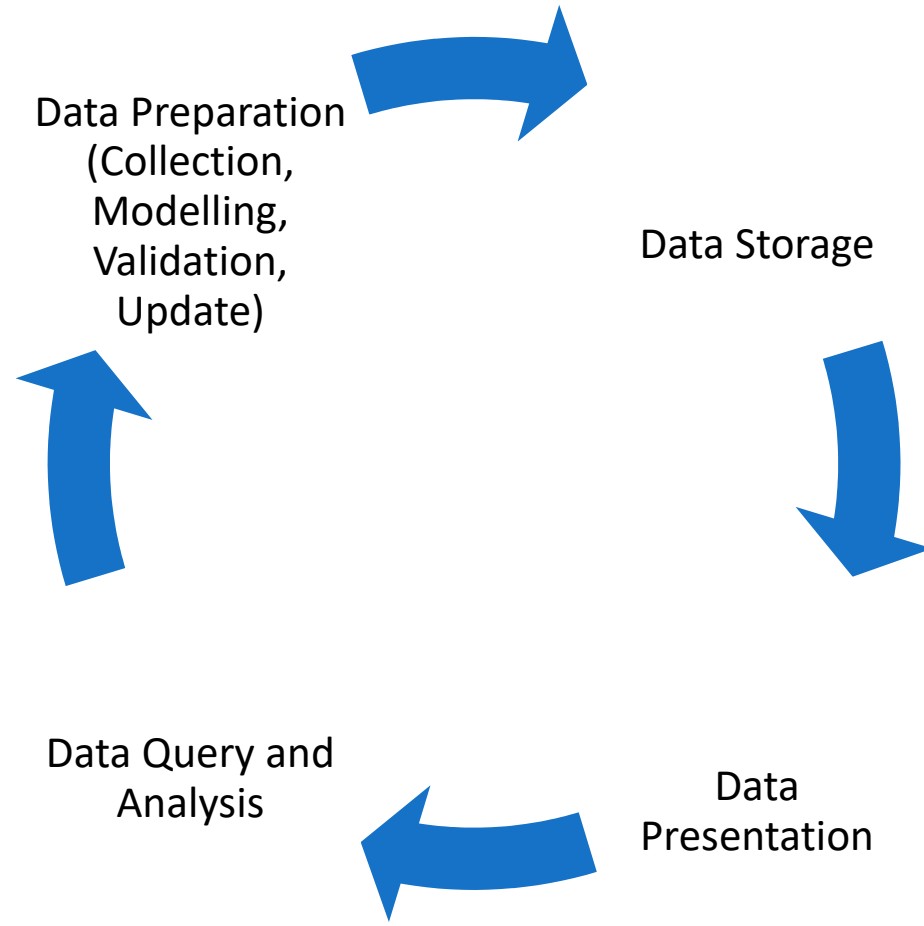

**Figure 1.** Digital data lifecycle.

During recent years, Building Information Models (BIM) have been identified as a good data source for a 3D digital cadastre [22–24]. Therefore, the integration of BIM into 3D digital cadastral systems is a research topic which has been explored by various scholars around the world. Most investigations in the field of a BIM-driven 3D digital cadastre have a limited focus on a very specific aspect of the lifecycle of 3D cadastral data. For example, some of these studies only consider BIM-based 3D data visualization techniques [25], while others focus on linking BIM/IFC with 3D cadastral data models [19,24]. In other words, these approaches are considered as a fragment of the entire lifecycle of 3D cadastral data,

which may not provide an end-to-end practical solution for implementing a BIM-based solution for a 3D digital cadastre within a specific jurisdiction.

The problem underpinning this research is that there is a lack of an efficient and streamlined approach for modelling, storing, visualizing, and querying BIM-based 3D cadastral data for multi-story buildings. Therefore, the aim of this study is to propose a new end-to-end approach for managing the lifecycle of 3D cadastral data using a BIM-based environment. To achieve this aim, the following steps have been undertaken:

1.  Model physical and legal components of a building subdivision plan in the open BIM-based data format, that is, IFC;
2.  Store the IFC file comprising 3D cadastral data in a 3D spatial database;
3.  Convert datasets from the 3D spatial database to Cesium 3D tiles for visualization purposes;
4.  Develop a 3D query system to query and retrieve the 3D cadastral data in a web environment.

In the next section, the existing research on different aspects of a 3D cadastral data lifecycle will be reviewed. This is followed by proposing a BIM-based approach developed for streamlining 3D cadastral data management in Section 3. Afterwards, the advantages of the proposed approach against the current 2D cadastral data lifecycle will be discussed in Section 4. Finally, the paper concludes with highlighting its major contribution and considerations for future research.

## 2. Related Works

The current practice for managing the lifecycle of cadastral data relies on 2D environments. The challenges of using 2D environments in complex situations have been widely recognized in the literature. As a result, different 3D-enabled solutions have been proposed to address a broad range of many issues in storing, manipulating, and representing cadastral aspects of complex built environments. In this section, we will provide an overview of the relevant literature from 3D cadastres in important stages of the 3D cadastral data lifecycle, namely, data preparation, data storage, data visualization, and data query.

### 2.1. 3D Cadastral Data Preparation

Preparation of 3D cadastral data relies on a comprehensive 3D data model and effective 3D data validation methods to create complete, accurate, and efficiently structured data. Three-dimensional data models are classified into three categories: legal, physical, and integrated models [26]. Legal 3D models provide 3D digital representation of ownership boundaries of private land and properties, common properties, easements, height and depth limitations, as well as ownership attributes. The prominent legal 3D models adopted for land and property management are the LADM at the international level [27] and ePlan model in Australia [28]. These models are adequate for registering the legal ownership of land and properties in a 3D digital environment, but the models' major issue is the inefficient communication of the location of ownership boundaries for non-technical people [29]. Physical models define semantic and spatial information about physical objects (such as buildings, roads, trees, tunnels, and bridges) in an urban environment. There are three prominent physical models: the CityGML [30] and IndoorGML [31], used by the geospatial industry, and Industry Foundation Classes (IFC), adopted widely by the architecture and construction industries [32]. Integrated models have emerged due to the need for linking legal and physical dimensions of urban environments [33]. Integrated representation of physical and legal models could help communicate and identify the location of ownership boundaries in a real-world context. Examples of integrated models include the integration of LADM and CityGML developed in China [34] and Poland [35], extending CityGML with legal information from the Dutch [36] and Turkish [37] jurisdiction perspectives, linking LADM and IndoorGML [38], connecting LADM and IFC [19], and the 3D cadastral data model, which extends the core cadastral data model by integrating urban features [33]. More recent integrations considered ontologies and linked data

approaches for developing integrated models. Linked data integrations used semantic web technologies and the Resource Description Framework (RDF) graphs to integrate IFC and CityGML standards [39,40]. Currently, there is limited research related to the development of integrated models using linked data to support a 3D digital cadastre.

Three-dimensional data capturing methods are significantly important for implementing 3D digital cadastral systems. There are three types of 3D cadastral data acquisition [41]:

1.  Range-based modelling, such as terrestrial laser scanning, LiDAR, and mobile mapping;
2.  Image-based modelling, such as terrestrial photogrammetry, high-resolution satellite imagery, and aerial photography [42];
3.  Integrated techniques: Integration of image-based and range-based modelling approaches provides a new solution that can source 3D data representing both the land parcel and the internal and external building details [43]. Land parcel extraction can be done using airborne imaging, while scanners can provide detailed 3D building data [44].

In a 3D digital cadastre, we can only capture physical objects, such as walls, doors, windows, fences, and so on for existing buildings. However, legal objects are virtual ones, which are defined by referencing to these physical entities. The above three methods only capture the reality, and various processes are required to classify these objects into meaningful entities. There are various studies about classifying captured data into semantic-based data models, such as CityGML [45] and IFC [46]. However, the cadastral objects need to be manually defined within these 3D models according to the position of legal boundaries.

While a 3D data model provides the structure for authoring and storing 3D cadastral data, 3D cadastral data can be prepared with good quality by considering intrinsic and extrinsic validation rules [47]. Intrinsic validation checks the data quality in each individual 3D cadastral object. Extrinsic validation checks the spatial relationships between a set of two or more 3D cadastral objects, and ensures that there is no gap or overlap between two different ownerships [48]. Various investigations identified the principles and rules for checking the spatial integrity and validity of the 3D cadastral data [49–51].

### 2.2. 3D Cadastral Data Storage

Storing 3D cadastral data can be done using two generic approaches: file or database. Both approaches rely on 3D cadastral data models for storing geometric and semantic information. Depending on the requirements, file-based or database storage can be adopted when creating and manipulating 3D cadastral data. For example, a surveyor may use a file for storing and lodging 3D cadastral data, while the land registry organization may use a 3D cadastral database for registering and updating 3D cadastral datasets. Most current BIM-based solutions for 3D cadastre rely on file-based approaches for storing 3D cadastral data [52,53]. However, there is limited research on the feasibility of 3D cadastral databases for storing cadastral data. The Reference [54] utilized the Oracle database as storage and the IFC Engine DLL as a converter tool to convert IFC files to database tables. object-oriented characteristics (Inheritance, Polymorphism, and Encapsulation) in the Oracle database were considered as a key point to select this software.

More recently, [21] proposed a new methodology for transforming 3D cadastral data into a BIM-based 3D spatial database. The proposed 3D cadastral database provides the basis for performing both semantic and spatial queries, which subsequently addresses the requirements for 3D cadastral data analysis. This study highlighted how constructing a 3D cadastral database relies on the approach used for capturing and authoring 3D cadastral data. Further studies looked at the use of ontologies, such as LADM OWL [55] and IfcOWL [56], for 3D data storage. Ontology-based 3D data storages have been considered for interlinking BIM and GIS domains to support urban applications, such as urban facility management [57]. There are few studies for overcoming semantic harmonization challenges when storing 3D cadastral data [58].

### 2.3. 3D Cadastral Data Visualization

Three-dimensional visualization of land and property data has been a research topic in 3D cadastres since the 2000s. Emerging new 3D digital technologies in computer graphics or improvements in hardware technology have stimulated researchers to design, develop, and test various visualization applications for the 3D cadastre. Several prototypes have been developed and tested for visualizing 3D cadastral data [25]. However, a limited number of these 3D visualization systems have been practically implemented in a few jurisdictions, such as the Netherlands [14]. Although there are similar activities in 3D digital cadastre prototype developments, validation of these prototypes is essential and fundamental before considering them as real-world 3D cadastral applications. Therefore, validation of 3D cadastral visualization prototypes is an important step towards developing end products, and the primary goal is validating the assumptions or methods [59].

### 2.4. 3D Cadastral Data Query

Efficient 3D data query mechanisms would facilitate the delivery of meaningful knowledge drawn from 3D cadastral data. It will unlock the value of cadastral data for a broad range of key stakeholders and catalyze them in decision-making for complex ownership situations. Three-dimensional cadastral data query methods can be classified into two categories: semantic and spatial. Semantic queries rely on the existing relationships between cadastral spaces, as well as their attributes [53]. These queries rely on SPARQL functions, which are specifically extended for BIM (i.e., BIMSPARQL [60,61]) and GIS (GeoSPARQL [62]) domains. Semantic queries are fast and easy to run when executing them in 3D cadastral systems. However, this type of query retrieves limited information required for decision-makers, since a limited number of semantic relationships between different cadastral spaces is defined in 3D digital models. On the other hand, spatial queries are more flexible and rely on a wide range of 3D spatial (directional, topological and proximity) operators to query 3D cadastral data [63]. A combination of 3D spatial operators would respond to many important queries related to complex ownership situations. However, compared to semantic queries, the speed of information retrieval in spatial queries could be slower, since these queries rely on the performance of complex 3D computations and spatial analyses.

All the above reviewed solutions for 3D cadastral data management are very limited in focus, such as 3D visualization and 3D data modelling. There is no study developing an integrated approach comprising the significant stages of the entire lifecycle of 3D cadastral data. These stages include 3D data preparation, 3D database storage, visualization, and querying 3D data. Therefore, this research aims to develop a new streamlined approach to recording, managing, and communicating cadastral data in a 3D digital environment.

## 3. Development of a New Approach for 3D Cadastral Data Management

The approach of this study is developed by integration of four main phases, as shown in Figure 2.

Each phase will be explained in detail in the following subsections. These phases have been implemented for a real-world case study building located in Melbourne, Australia. The case study building includes a wide range of legal spaces and boundaries, which makes it a suitable example for implementing the proposed approach in order to manage the lifecycle of 3D cadastral data.

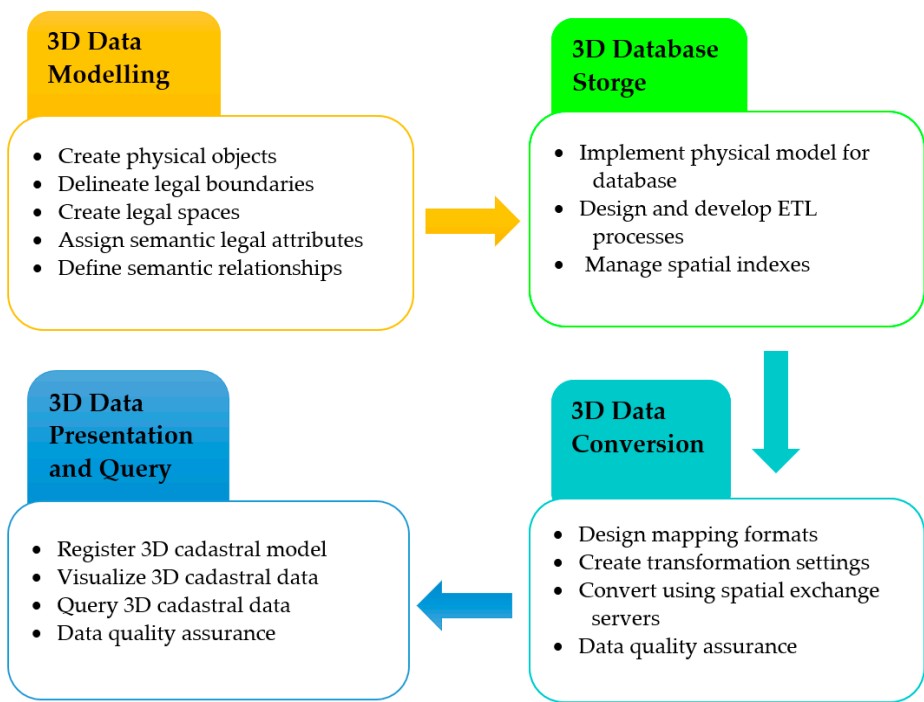

**Figure 2.** Proposed approach for 3D cadastral data management.

### 3.1. 3D Cadastral Data Modelling and Preparation

To model and prepare 3D cadastral data, Autodesk Revit was used as a BIM authoring tool. The following steps were undertaken to create 3D cadastral data in the Revit environment, which is subsequently exported to IFC format:

1. Modelling physical objects: In this step, basic architectural elements, which define the legal boundaries, are placed in the BIM model. These elements include floors, ceilings, doors, windows, columns, and any physical component used for defining legal boundaries. Further details about creating physical objects in Revit can be found on the Autodesk website [64].

2. Delineating legal boundaries: This activity includes defining various types of legal boundaries of ownership spaces in Revit. In the context of this study, there are three common types of building boundaries defined by referencing the interior, median, and exterior faces of architectural elements (see Figure 3). In addition, boundaries defined by surveying measurements, that is, bearing and distance, were also delineated in Revit. A detailed explanation of modelling legal boundaries in the BIM environment has been provided in [65].

3. Creating legal spaces: After delineating boundaries in Revit, volumetric legal spaces were created based on the defined boundaries. These spaces include apartment units, storage areas, parking lots, and communal areas, including corridors, lobbies, and stairs.

4. Assigning legal attributes: For any building component (such as a wall, door, window, and space), Revit provides a predefined set of parameters (attributes) which are mainly for architectural and engineering purposes. However, there exists a mechanism in Revit to add new parameters (such as legal attributes) for the vast majority of building elements. We used this capability to add required attributes to legal spaces. When exporting the final BIM model to the IFC format, these attributes were shown as property sets in the IFC file.

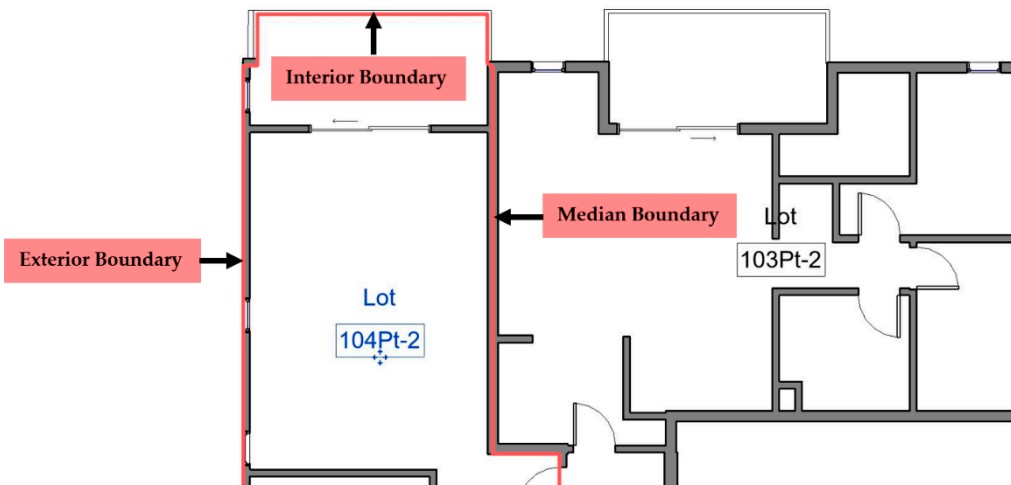

**Figure 3.** Examples of interior, median, and exterior boundaries delineated within a BIM authoring tool.

### 3.2. Storage of IFC Files in 3D Spatial Database

The main purpose of this step is storing the IFC file in a spatial database, which is based on Oracle technology in this study. To undertake this step, the FME software was used for the 3D data storage process. What FME needs for this process is an IFC file, as well as a connection to the user's account within a pre-existing Oracle database.

To make a link between the FME and Oracle database, a new connection must be created in FME to enable the connectivity of the Oracle Spatial Object. Once the connection between the FME and Oracle spatial database is established, then the IFC file can be added to the FME workspace as a Reader. The default measurement unit for height in BIM authoring tools is normally millimeters, and FME does not have any capability to change the measurement unit. Therefore, the IFC file should be exported in a meter unit; otherwise, the 3D IFC model will be collapsed during the visualization stage.

The IFC file includes a large number of entities. To get the required data for a 3D digital cadastre, the parameter of the Reader should be set as "Building Elements with Hierarchy". In addition, the "Coordinate System" should be set as "ESPG:4238" to get the geospatial coordinates; otherwise, the coordinate system will be stored in a local format in the 3D spatial database.

Once the required features for the 3D digital cadastre from the IFC file are read by FME, a conversion model should be designed to map the IFC 3D entities into 3D spatial objects in the Oracle database. The conversion model provided in Figure 4 shows this mapping.

As shown in Figure 4, the conversion model consists of three main components: Reader, Transformer, and Writer. Reader is the origin or data source for the conversion. In this diagram, IFC entities are the Reader. Transformer is the action or setting to convert the source data to the destination format. Any translation, restriction, filter, or setting should be defined in Transformer, and this component plays an intermediary role between Reader and Writer. Writer is the destination of the data exchange or transformation. In this case, Oracle 3D spatial objects are considered as the components for the Writer.

In this study, all the unnecessary entities have been removed from the initial IFC file loaded into FME, as they either do not include attribute/spatial data or are not required for 3D cadastral purposes. For example, Beam, FlowController, FlowFitting, Footing, FurnishingElement, and EnergyConversionDevice have been eliminated.

The next step is determination of Oracle 3D spatial objects in the destination database. For any IFC entity, one database table can be considered. This one-to-one mapping of IFC entities to database tables is the approach utilized in this study. By default, all database tables in FME are configured as 2D spatial tables. In this study, we configured those tables as 3D by setting the Z coordinate parameter. The other consideration is that the proposed column names in database tables will be the same as attribute names in IFC entities, which

may cause issues in creating Oracle tables due to the limitation on the length of the column name. This issue can be addressed by updating the corresponding attribute name in tables.

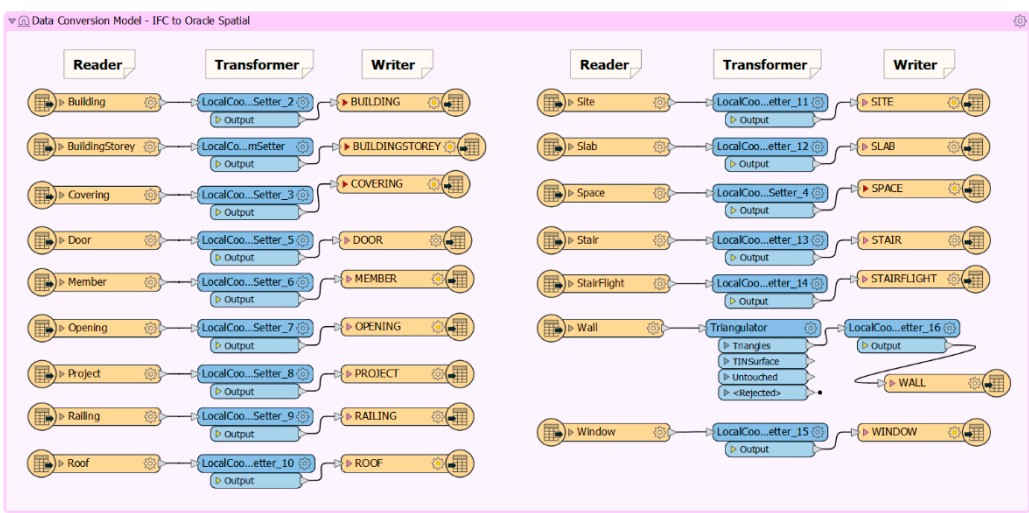

**Figure 4.** A conversion model developed in FME for mapping IFC data into the Oracle 3D database.

In the process of mapping IFC entities to Oracle spatial tables, two main challenges were identified: coordinate system transformation and wall features conversion. Although the input IFC file is in the Map Grid of Australia (MGA) coordinate system, the coordinates of each entity are stored in local values inside the IFC. In IFC viewers, by using the IfcSite element, the local values will be automatically transformed to MGA coordinates based on the IfcSite's MGA coordinates, which define the origin (X = 0, Y = 0) for other IFC entities. FME is not capable of performing this transformation automatically based on the IfcSite. In a default transformation, all the coordinates of IFC entities will be transformed to Oracle in local values. To address this challenge, an FME transformer is defined to convert the coordinates from a local to MGA geospatial coordinate system. This transformer is called the LocalCoordinateSystemSetter and is placed between IFC entities and their corresponding database table using the configuration shown in Figure 5.

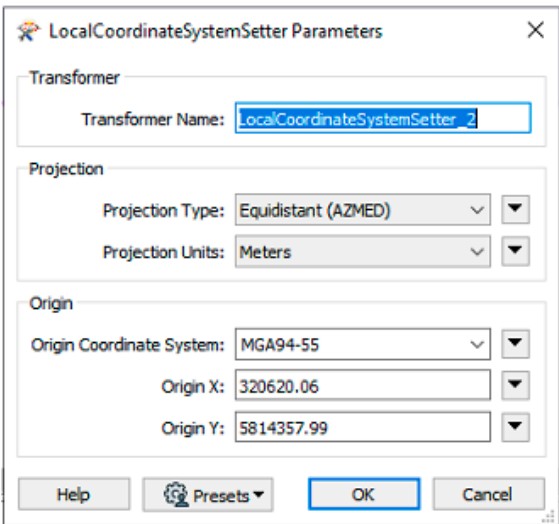

**Figure 5.** Configuration of coordinate system transformation for IFC data in FME.

The X and Y coordinates in the configuration are the same as the coordinates of the IfcSite entity in the IFC file in the MGA coordinates system. This transformer is utilized to transform all the entities in IFC to the Oracle spatial with geo-referenced values.

The other challenge was that FME converts the wall including windows to an invalid solid geometry in the Oracle database. This challenge could not be identified until the database tables were transformed to Cesium 3D tile sets. To overcome this issue, a new FME transformer called Triangulator was defined between the IfcWall entity and its corresponding table in the database. The transformer splits the wall geometry into triangular units similar to a mesh, and subsequently, all the IfcWall entities are converted to valid solid geometries in the 3D spatial database.

*3.3. Development of an IFC to Cesium 3D Tiles Converter*

Once the IFC entities were stored in the database, 3D data associated with each entity could be edited, queried, visualized, and analyzed in the database. This is the key benefit of storing 3D cadastral data in a database, as compared with flat file storage. In addition, complex 3D spatial analyses and computations can be performed on 3D cadastral data.

In this phase, the IFC data stored in the database were converted to the Cesium 3D tile set to be represented in a web-based 3D visualization platform. For converting the spatial database to Cesium 3D tile sets, database tables were selected as a Reader in FME and using the AggregateFilter transformer, the tile set was created as a unique object. AggregateFilter makes a unique object from the combination of all entities in database tables. As a result, one tile set was created. The alternative approach is creating one tile set per single entity, which was found as an inefficient approach in this study since entities could not be visualized in a fully integrated manner.

Figure 6 shows that a many-to-one relationship between database tables and tile sets is used for performing the conversion. It should be noted that the database schema name is EPLAN, and that is why the Reader objects are shown with the EPLAN prefix in Figure 6. The only required setting for the Oracle spatial object as a Reader is to enable the "Read 3D Polygons as Faces" parameter. Through setting this parameter, all the 3D polygons in database tables are read as 3D face geometries.

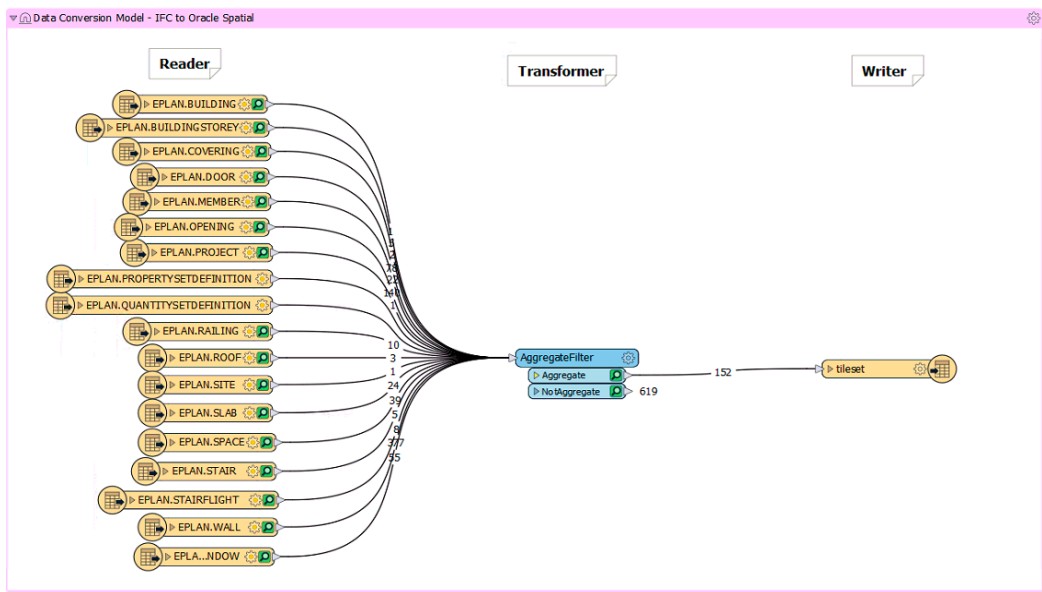

**Figure 6.** Converting 3D spatial objects from the Oracle database into the Cesium 3D tile set.

Once all the settings and mappings were completed, the conversion process was automatically run within FME, and two files were created: a tile set JSON file and a datafile associated to it. The JSON file reads the datafile and represents it in 3D format, as shown in Figure 7. This Figure shows an integrated representation of 3D physical elements and legal spaces for the case-study building.

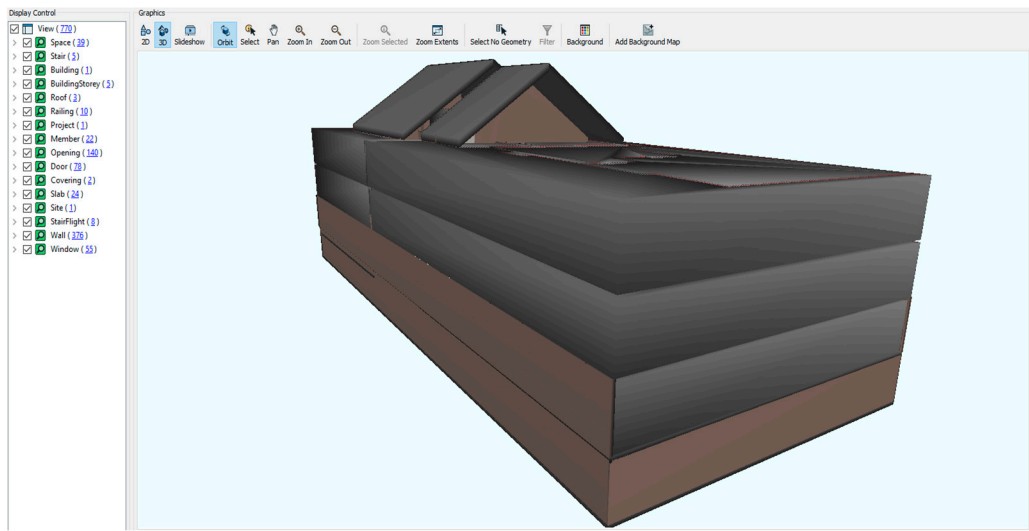

**Figure 7.** Visualization of physical elements and legal spaces converted to the Cesium 3D tile set.

### 3.4. 3D Cadastral Data Query & Visualisation System

Once the generated 3D tile file is published in a web server (e.g., Apache, Jetty), it can be directly consumed and visualized by any CesiumJS-based web application. In this study, a 3D prototype system for 3D data query and visualization was implemented as an integrated environment for retrieving the 3D cadastral data intuitively. To register a new 3D cadastral model, the system requires parameters including the address, coordinates, and 3D tile URL, and then the 3D model is ready for visualization.

The implemented prototype system offers different tools to visualize and query 3D cadastral models. The system view contains two parts: the control panel is on the left side, and the visualization and query component sits on the right side, which offers a full 3D view and interaction with the model. By default, the legal spaces (which are defined as IfcSpace) are displayed with semi-transparency. In Figure 8, the green blocks represent "Common Property", and the blue blocks are "Lots".

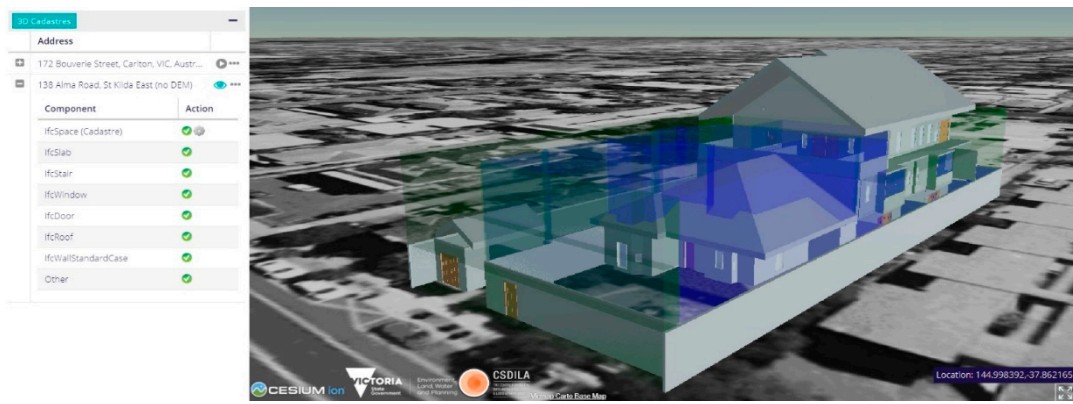

**Figure 8.** Visualization of legal spaces for common property and lots within the prototype system.

Besides the IfcSpace component, the system also has separate control (by ticking and unticking the green ticks) of the visibility of other IFC entities, such as IfcRoof, IfcWallStandardCase, and IfcWindow. For example, the top-left image of Figure 9 shows all the building components except IfcSpace; the top-right image shows a combination of IfcSpace, IfcSlab and IfcWallSatandarCase; the bottom-left image reveals more interior details by removing the IfcRoof components and the bottom-right image only illustrates the IfcSlab,

IfcDoor and IfcStair. This is found useful to visually check the space relationships between a cadastral model and other building components.

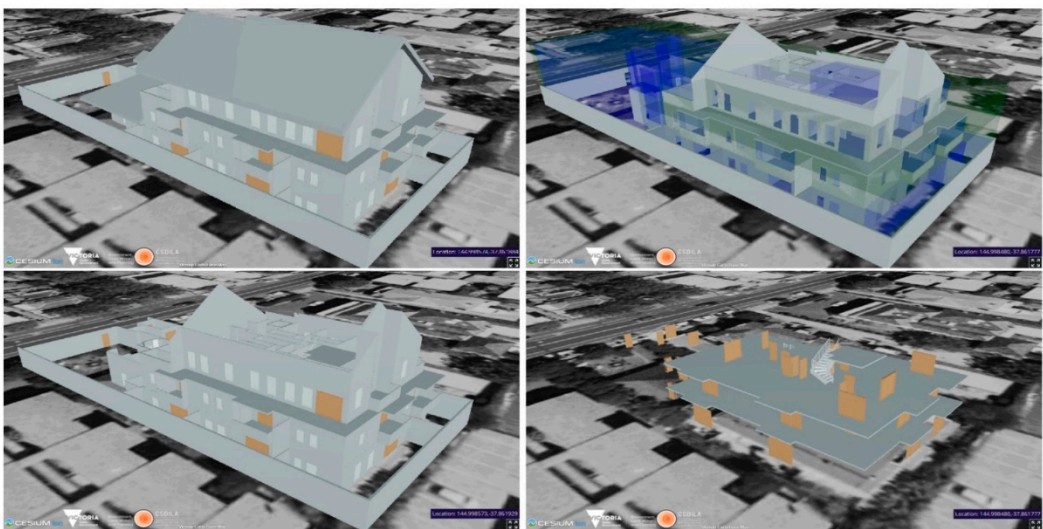

**Figure 9.** Visualization of the 3D cadastral model within the prototype system.

To query the cadastral data, the system provides a comprehensive user interface to build search criteria. The query will be sent to the back-end database, which will return the "id" of all found records for visualization. For instance, in Figure 10, a query for all lots located on levels 1 and 2 with mortgagee 'NAB' is processed, and three results are returned and highlighted in red. The users can click on any lot, which will be highlighted in green, to check the detailed information.

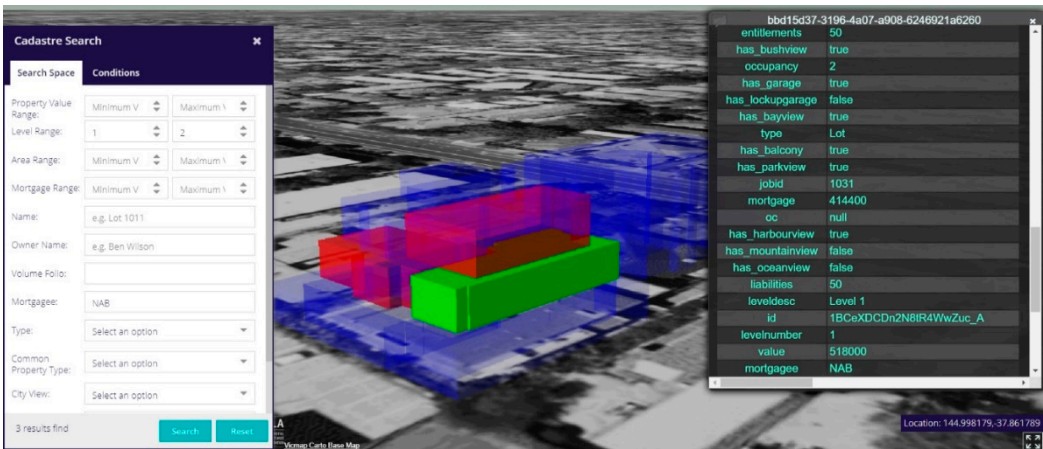

**Figure 10.** Querying various 3D legal spaces in the 3D cadastral model.

It should be noted that cadastral data can also be stored within the 3D tile files by defining and populating the attributes inside its 'batch table'. The benefit will be that all queries can be parsed and handled on the front-end without database connection. However, this design has more drawbacks than benefits. First of all, cadastral data are not static, and whenever any update occurs, the entire 3D tile needs to be regenerated and deployed to reflect the changes. Secondly, it is not a scalable design and will become inefficient when facing more complex queries and a large number of cadastral components. Thirdly, when storing cadastral attributes inside a 'batch table', it actually forms isolated datasets, which makes searching cadastral data a challenging task. According to all these drawbacks, the developed system uses a back-end database for the cadastral data query.

## 4. Discussion

The use of a 2D-based environment for data curation in multi-story buildings is highly entrenched in current cadastral processes [66]. This long-established paradigm encounters significant challenges when recording, storing, and managing spatially interconnected complexities of legal spaces. Therefore, this study aimed to develop a new approach for managing the lifecycle of cadastral data in 3D environments. The proposed approach provides significant benefits when creating, storing, and communicating cadastral data in complex multi-story buildings. Here, we will compare the proposed 3D approach with current 2D-based practices in important stages of processing cadastral data:

- Data creation: In the current 2D practices, CAD drawings are typically used for creating cadastral data related to the subdivision of multi-story buildings. The creation of these drawings is a significantly challenging task when a building has complex architecture. The land surveyor needs to create many floor plans, cross-sections, and isometric views to capture the full extent of all legal spaces. During this process, only the geometry of legal spaces is created inside the CAD drawings, and there is no semantic information associated with these legal spaces. In addition, while the surveyors use architectural plans as a basis for delineating legal boundaries in buildings, physical elements are eliminated from subdivision drawings. Our proposed approach for 3D cadastral data creation in a BIM-based environment provides the ability to create a fully integrated 3D geometry of legal spaces with a rich amount of semantic attributes and relationships. Furthermore, physical elements are not removed during our proposed 3D data creation process.

- Data storage: Two-dimensional CAD drawings are stored in a file format. The file-based data storage approach is not efficient when storing cadastral data about high-rise building subdivisions. The CAD file size will be significantly increased in high-rise buildings, since many 2D floors, cross-section diagrams, and isometric views will be created inside the file. This will also make it challenging to record and track the changes in cadastral data. The file-based approach also does not enable users to edit and update the data. This would impede the collaboration in complex projects, resulting in significant delays and costs [67]. To address these issues, our proposed approach relied on a 3D spatial database to store 3D digital cadastral data for multi-story buildings. The implemented 3D spatial database provides different types of accesses for users and enables tracking and editing of cadastral data in complex projects. As a result, collaboration among different stakeholders and players involved in the lifecycle of 3D cadastral data will be potentially fostered.

- Data conversion: In the current land registration process, the surveyors should provide a PDF version of subdivision plans for multi-story buildings. Although the conversion from CAD to PDF format makes subdivision plans more human-readable, important data elements are not machine-readable in PDF format, which hinders its potential future reuse. On the other hand, the proposed 3D data conversion approach ensures that all required data elements coming from BIM/IFC data are stored in their corresponding tables within the 3D spatial database (see Figure 4). The conversion at the data level, rather than at a purely presentation level, would potentially provide further benefits and applications beyond 3D land registration purposes.

- Data communication: The use of 2D PDF files and CAD drawings limits the value of cadastral data related to complex multi-story developments. Using a 2D environment to capture the 3D reality does not provide an accurate understanding of the legal spaces and associated boundaries. In addition, 2D PDF plans provide a static representation of legal spaces, which is a significant barrier to the interaction of users with cadastral data. This will make it difficult for users to search, find, and retrieve the required cadastral data in multi-story buildings. In response to these limitations of 2D data communication, the implemented 3D data query and visualization system provides an interactive and dynamic 3D digital environment for users to query, search, and identify legal spaces, boundaries, and associated ownership attributes in complex buildings.

This 3D system provides a visually rich representation of legal and physical objects in a common data environment, enabling the users to link the legal boundaries and spaces to the physical reality. This would potentially unlock significant benefits for a wide array of urban applications that rely on 3D cadastral data. Therefore, the implemented 3D system provides a basis for developing various 3D spatial data analyses which can tap into the valuable knowledge driven from 3D digital cadastral data.

The above points demonstrate the clear advantages of a 3D approach for managing cadastral data in urban built areas. The suggested approach contributes to significant improvements in the present 2D-based practice of managing land and property data by addressing new 3D digital methods to acquire, manage, and disseminate cadastral data. Many parties involved in land and property development will be able to access and use 3D cadastral data. Compared to existing studies which are focused on a specific phase of the digital lifecycle, such as 3D visualization or 3D data modelling, the proposed approach in this study covers the entire cadastral data lifecycle. The main benefit is providing practical guidelines required for implementing a 3D digital cadastre. In other words, it shows how 3D cadastral data should be created, how it should be stored in a 3D spatial database, and how it can be represented and queried within a 3D digital environment. However, the realization of this approach in the current process is a major and challenging upgrade for different jurisdictions around the world. It requires a robust roadmap for undertaking all the required steps to shift from a 2D to 3D cadastral data lifecycle.

## 5. Conclusions and Future Work

This study investigated the potential of a new approach for streamlining the 3D cadastral data lifecycle in order to mitigate the current challenges associated with data preparation, storage, management, and communication in cadastral systems. This work mainly contributes to the development of a practical and simplified solution for managing the lifecycle of 3D cadastral data in current practices. The identified steps for capturing, storing, and communicating 3D cadastral data are critically important in transforming the current 2D systems to 3D digital information systems.

There are still significant research challenges that should be addressed in future works. Through the approach developed in this study, editing and updating the 3D cadastral data can only be done directly in the 3D spatial database. However, for better user experience, the 3D visualization system should provide the ability to edit and update 3D cadastral data through its user interface. It means that the system should write the data back to the 3D spatial database. This will also be possible by connecting BIM authoring tools, such as Revit, to the 3D spatial database. The bi-directional connection between BIM authoring tools and 3D spatial databases will affect 3D data visualization and query services. As the user changes the BIM model, 3D data visualized and queried from the 3D spatial database will be updated accordingly. In addition, the proposed 3D spatial database schema only considered legal and physical objects. In future research, this schema can be extended with survey data elements, such as traverse lines, control points, and reference and traverse points to unify 3D cadastral data with survey data in one place. The implemented 3D data visualization and query system can be tested further by measuring the query time and visualization speed when interacting with a range of legal spaces, as well as boundaries and ownership attributes associated with these spaces. In addition to the technical challenges, the eventual costs associated with the technical implementation of the 3D digital cadastre needs to be investigated in further studies.

**Author Contributions:** Conceptualization, H.O.; Data curation, D.S., F.B. and Y.C.; Funding acquisition, A.R.; Investigation, B.A., F.B., Y.C. and H.O.; Methodology, H.O. and B.A.; Project administration, A.R.; Software, Y.C.; Supervision, A.R.; Validation, F.B.; Writing, H.O., B.A., Y.C., D.S. and F.B. All authors have read and agreed to the published version of the manuscript.

**Funding:** This research was partially funded by Land Use Victoria, within the Department of Environment, Land, Water and Planning (DELWP).

**Institutional Review Board Statement:** Not applicable.

**Informed Consent Statement:** Not applicable.

**Data Availability Statement:** Restrictions apply to the availability of the data used in this study. Data was obtained from a third party and is available from the authors with the permission of the third party.

**Acknowledgments:** The authors acknowledge the support of Centre for Spatial Data Infrastructures and Land Administration, The University of Melbourne, and Land Use Victoria within the Department of Environment, Land, Water and Planning (DELWP). The authors emphasize that the views expressed in this article are the authors' alone.

**Conflicts of Interest:** The authors declare no conflict of interest.

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
