# Peer review of "A Proposal for Streamlining 3D Digital Cadastral Data Lifecycle"

_land, doi:10.3390/land10060642_

Round 1

Reviewer 1 Report

Dear colleagues, thank you for your contribution on "A Proposal for Streamlining 3D Digital Cadastral Data Lifecycle". Indeed the topic of 3D digital cadastral data lifecycle needs more attention, especially with recent developments in recording, modelling, storage and analysis methods. 

The contribution touches the most important aspects of the data lifecycle: data creation, data storage, data conversion, data communication. 
In all those aspects new perspectives and methods have been created recently, which are missing in this contribution. For example "linking" and "semantic" has been mentioned in section 2, but its perspective to graph representations (in the database) and linked data integration are not even referenced. The same is true for data communication and data storage. 

In 2.1 the 3D cadastral data acquisition is categorized in three classes, which obviously do not cover modern data integration and processing methodology. For example a common splitting of geographic and thematic information, a semantic structuring of the legal and physical entities as well as the role of point-based classification schemas should be embedded in the argumentation and modelling of 3d cadstral data acquisition. 

Figure 2 proposes an approach for 3D cadastral data management: unfortuneately the semantic dimension and data-integration is completely missing in this data managment workflow.

We would expect a more inclusive point of view from this expert domain of cadastre, as the cadastre and 3D cadastre is part of a common data science processing methodology. 

Author Response

Dear Reviewer,

Thanks for your invaluable feedback. Please see the table below for our responses to your comments.

Reviewer’s feedback

Authors’ response

Dear colleagues, thank you for your contribution on "A Proposal for Streamlining 3D Digital Cadastral Data Lifecycle". Indeed the topic of 3D digital cadastral data lifecycle needs more attention, especially with recent developments in recording, modelling, storage and analysis methods.

Thanks for your feedback. By considering your specific comments, we have addressed all the recent developments in various aspects of 3D digital cadastral data lifecycle.

The contribution touches the most important aspects of the data lifecycle: data creation, data storage, data conversion, data communication.

In all those aspects new perspectives and methods have been created recently, which are missing in this contribution. For example "linking" and "semantic" has been mentioned in section 2, but its perspective to graph representations (in the database) and linked data integration are not even referenced. The same is true for data communication and data storage.

In response to this comment, the following statements been added to Section 2.1:

More recent integrations considered ontologies and linked data approaches for developing integrated models. Linked data integrations used semantic web technologies and Resource Description Framework (RDF) graphs to integrate IFC and CityGML standards [35,36]. However, there is limited research related to the development of integrated models using linked data to support requirements of 3D digital cadastre.

In addition, the following statements relevant to data storage is added to Section 2.2:

Further studies looked at the use of ontologies, e.g. LADM OWL [48] and IfcOWL [49], for 3D data storage. Ontology-based 3D data storages have been considered for interlinking BIM and GIS domains to support urban applications such as urban facility management [50]. There are few studies for overcoming semantic harmonization challenges when storing 3D cadastral data [51].

Finally, the following statements relevant to data communication is added in Section 2.4:

These queries rely on SPARQL functions which are specifically extended for BIM (i.e. BIMSPARQL [53,54]) and GIS (GeoSPARQL [55]) domains.

In 2.1 the 3D cadastral data acquisition is categorized in three classes, which obviously do not cover modern data integration and processing methodology. For example a common splitting of geographic and thematic information, a semantic structuring of the legal and physical entities as well as the role of point-based classification schemas should be embedded in the argumentation and modelling of 3d cadstral data acquisition.

In response to this comment, the following statements have been added to Section 2.1:

In 3D digital cadastre, we only can capture physical objects such as walls, doors, windows, fences and so on for existing buildings. However, legal objects are virtual objects which are defined by referencing to these physical entities. The above methods only capture the reality and various processes are required to classify these objects into meaningful entities. There are various studies about classifying captured data into semantic-based data models such as CityGML and IFC. However, the cadastral objects need to be manually defined within these 3D models according to the position of legal boundaries.

Figure 2 proposes an approach for 3D cadastral data management: unfortunately the semantic dimension and data-integration is completely missing in this data management workflow.

In response to this comment, we clarify the data integration was done in the first stage, which is 3D data modelling. In fact, the semantic dimension (including semantic attributes and relationships) associated with legal and physical objects have been considered in this stage. Therefore, Figure 2 has been updated accordingly and relevant explanation is added to Section 3.1

We would expect a more inclusive point of view from this expert domain of cadastre, as the cadastre and 3D cadastre is part of a common data science processing methodology.

By considering your comments and new changes in the revised manuscript, we provided more inclusive point of view in the Discussion section.

Reviewer 2 Report

land-1240547

The manuscript “A Proposal for Streamlining 3D Digital Cadastral Data Lifecycle” addresses an interesting and up-to-date subject, which adhere to Land journal policies.

The manuscript contains interesting results and presents a good 3D cadastre application. In addition, the work is well-conceived and written, so that I did not identify deficiencies or shortcomings that would require major revisions or improvements.

In my opinion, the manuscript must be minor improved before publication with the following:

  • Abstract improvement, with more Results information
  • Additional information regarding the software used
  • If possible, some aspects regarding eventual costs when implementing such 3D cadastre
  • I would recommend a traditional structure with chapter 2. M&M and 3. Results

Author Response

Dear Reviewer,

Thanks for your invaluable feedback. Please see attached for our responses to your comments.

Reviewer 3 Report

The paper outlines a process that leads from original 3D surveys to data models for storage, visualization, and analysis. The authors show the different transformations necessary and this will help in the future to focus on the single steps and improve them.

In section 5, the authors mention an important aspect that is still missing: "... the 3D visualization system should provide the ability to edit and update 3D cadastral data through its user interface." I agree with that, but would like to make a more general request: Why stop at the 3D spatial database and not feed the change also back into the BIM model? It would be best, if I can edit the data on each level and the changes are propagated automatically in both directions of processing chain. However, this would require the inverse operations for the transformations described in this paper.

In the description of Fig 8 the authors explain the possibilities of the UI. Is it also possible to remove everything above a specific floor? If I want to see the structure of the second floor, the easiest approach would be removing the 3rd, 4th, etc. floor and the roof. Does the data model allow this? If not, such an extension could also be interesting for future work.

There seems to be a problem with the width of several figures (2, 3, 5, 6, 7, 8, and 9). Try to make them fit the page limits better without reducing the font size too much. In case of Figures 2 and 3, the rearrangement of the elements may be a solution.

Other than that I only have small comments:
- Line 36/37, "The complexities can arise from technical, institutional, or legal aspects of": What about combinations? Are they possible?

- Figure 1 and the description in the text (lines 39-57): This is a general model for working with updated data. It does not really matter that this data is 3D. The concept applies to all kinds of data 1D to nD and is as well not restricted to the cadastral domain. The cycle for the conventional 2D cadastre is the same. I suggest to drop the term "3D" in the steps and just apply the general concept of a series of steps on the application of 3D cadastre. Thus I would also change the caption to (I do not see the cadastre in the figure - why should it be mentioned in the caption?):
Figure 1: Digital data lifecycle

- line 118, "and 3D cadastral data model [29].": This last element somehow does not fit into the list since it does not explain what the integrative part in the referenced paper is. The paper should be listed here but could be specified that it integrates urban features in the core cadastral model.

- line 123: How do cadastral and architectural plans fit in the category "image-based modelling"? Is that really image-based? They are vector geometries if the data are available in digital form. The remainder of the text suggests that you think of these plans as PDF documents but that should be made specific.

- line 212-213, interior, median and exterior face": What is meant by interior and exterior? Could you explain that using a wall between neighboring apartments as an example? I cannot see in this case what an interior or exterior face would be.

- line 235, "stablished": Did you mean "established"? The dictionary tells me that "to stablish" is a correct but outdated verb, so it may either be on purpose or it is a typo ...

- line 258: You state that "all the unnecessary entities have been removed" but the windows are kept as elements of the data model. Why do we need information on windows in the 3D cadastre? Do we need them later to check building regulations?

Author Response

(The authors gave the same response as above.)

Reviewer 4 Report

Paper tackles an interesting problem in development of 3D cadastral systems. It describes a method how to create a 3D building information that can be stored in a 3D spatial database and how this can be visualised using 3D tiles.

Figure 1 claims to illustrate the current technical approach how 3D cadastres are created. I am not personally an expert of how these are in practice implemented but from a general spatial infrastructure point of view the process seems strange. The Figure claims the visualisation would be a prerequisite for making an spatial analysis or query which is definetly wrong.  There is no connection between those.  There is no reference how this figure is related to the current 3D cadastre implementation and no argument why visualisation would be needed as a step.

When Figure 2 is examined a further explanation of this step call visualisation can be found. It is not actually a visualisation step, it is creation of the 3D spatial presentation. 

The question then arises why there is a need to create two separate steps. First create a 3D storage that is not actual a 3D spatial representation and then an additional step to create a 3D spatial representation that can be used.  The reason probably is in the difference of BIM models and spatial data models. The first step is actual storing the BIM model in spatial database and the next step is transforming this BIM model to spatial model that can be analysed using spatial methods.

I would suggest to change the Figures 1 and 2 and correct the text accordingly.  

The current paper suggest one quality assurance step is needed but in reality there is two quality assurance steps needed.

The first quality assurance is needed when BIM model is stored in the spatial database. Actually the writers already have noted a reason for this when the note that FME could not found out some errors in walls when writing this.  In reality at least in Finland the BIM/IFC models would be coming from architectural bureaus to the municipalties which the would store the models when a building permit would be then given. Currently this BIM model could also be stored as a 3D Cadastre and stored as a file. As writers suggest there are issues with this approach as it is not real useful and can not be analysed.  

The next step is to create a 3D spatial representation which enables spatial analysis and visualisation. This requires also a transformation and quality assurance as suggested by the writers. 

The discussion part is a bit strange as the writers suggest improvements in current 3D cadastres but the main discussion is based on why 3D cadastre is need compared to 3D cadastres. I would suggest to focus on why this improvement is need and discuss the benefits of the approach suggested.

A small comment on the 3D tiles. Probably it could be considered for visualition purposes but in reality the 3D spatial storage could be done without using 3D tiles. If the authors would follow my suggestion and change the approach slightly there a spatial modelling would be better done with our 3Dtiles. Anyway 3Dtiles is not a correct approach for spatial analysis.

If this rather minor changes are done then this paper is an excellent paper. 

Author Response

Dear Reviewer,

Thanks for your valuable feedback. Please see the table below for our responses to your comments.

Reviewer’s feedback

Authors’ response

Paper tackles an interesting problem in development of 3D cadastral systems. It describes a method how to create a 3D building information that can be stored in a 3D spatial database and how this can be visualised using 3D tiles.

Thanks for your feedback.

Figure 1 claims to illustrate the current technical approach how 3D cadastres are created. I am not personally an expert of how these are in practice implemented but from a general spatial infrastructure point of view the process seems strange. The Figure claims the visualisation would be a prerequisite for making an spatial analysis or query which is definitely wrong.  There is no connection between those.  There is no reference how this figure is related to the current 3D cadastre implementation and no argument why visualisation would be needed as a step. When Figure 2 is examined a further explanation of this step call visualisation can be found. It is not actually a visualisation step, it is creation of the 3D spatial presentation.

As advised by the reviewer, Figure 1 and the relevant description have been updated accordingly. In particular, the “visualisation” has been changed to “presentation”.

The question then arises why there is a need to create two separate steps. First create a 3D storage that is not actual a 3D spatial representation and then an additional step to create a 3D spatial representation that can be used.  The reason probably is in the difference of BIM models and spatial data models. The first step is actual storing the BIM model in spatial database and the next step is transforming this BIM model to spatial model that can be analysed using spatial methods.

As advised by the reviewer, we have updated Figure 2 and the relevant description.

I would suggest to change the Figures 1 and 2 and correct the text accordingly. 

As advised by the reviewers, Figures 1 and 2 have been updated and the text has been corrected.

The current paper suggest one quality assurance step is needed but in reality there is two quality assurance steps needed.

The first quality assurance is needed when BIM model is stored in the spatial database. Actually the writers already have noted a reason for this when the note that FME could not found out some errors in walls when writing this.  In reality at least in Finland the BIM/IFC models would be coming from architectural bureaus to the municipalities which the would store the models when a building permit would be then given. Currently this BIM model could also be stored as a 3D Cadastre and stored as a file. As writers suggest there are issues with this approach as it is not real useful and cannot be analysed. 

The next step is to create a 3D spatial representation which enables spatial analysis and visualisation. This requires also a transformation and quality assurance as suggested by the writers.

As advised by the reviewer, we’ve added the QA step as part of the Data Presentation and Query phase in Fig 2.

The discussion part is a bit strange as the writers suggest improvements in current 3D cadastres but the main discussion is based on why 3D cadastre is need compared to 3D cadastres. I would suggest to focus on why this improvement is need and discuss the benefits of the approach suggested.

In response to this comment, the following paragraph has been added to the Discussion section:

The suggested approach contributes to significant improvements in the present 2D-based practice of managing land and property data by addressing new 3D digital methods to acquire, manage, and disseminate cadastral data. Many parties involved in land and property development will be able to access and use 3D cadastral data. Compared to existing studies which are focused on a specific phase of digital lifecycle, e.g. 3D visualisation or 3D data modelling, the proposed approach in this study covers the entire cadastral data lifecycle. The main benefit is providing practical guidelines required for implementing 3D digital cadastre. In other words, it shows how 3D cadastral data should be created, how it should be stored in a 3D spatial database, and how it can be represented and queried within a 3D digital environment. 

A small comment on the 3D tiles. Probably it could be considered for visualization purposes but in reality the 3D spatial storage could be done without using 3D tiles. If the authors would follow my suggestion and change the approach slightly there a spatial modelling would be better done with our 3Dtiles. Anyway 3Dtiles is not a correct approach for spatial analysis.

If this rather minor changes are done then this paper is an excellent paper.

In response to this comment, we clarify that 3D tiles are used for only visualizing 3D cadastral data. The storage of 3D cadastral data has been in the 3D spatial database, which is also used for performing spatial analysis.

Round 2

Reviewer 4 Report

Figure 1 still in my mind does not represent general digital data lifecycle process and could be clarified. The general data lifecycle process is create-storage-use-share-archive-destroy type. 

The Figure 2 now is much better and clearly indicates some challenges in the the management of 3D cadastral data.